# The evolutionary drivers and correlates of viral host jumps

Cedric C. S. Tan [1,2] ✉, Lucy van Dorp [1,3] & Francois Balloux [1,3]

Most emerging and re-emerging infectious diseases stem from viruses that naturally circulate in non-human vertebrates. When these viruses cross over into humans, they can cause disease outbreaks, epidemics and pandemics. While zoonotic host jumps have been extensively studied from an ecological perspective, little attention has gone into characterizing the evolutionary drivers and correlates underlying these events. To address this gap, we harnessed the entirety of publicly available viral genomic data, employing a comprehensive suite of network and phylogenetic analyses to investigate the evolutionary mechanisms underpinning recent viral host jumps. Surprisingly, we find that humans are as much a source as a sink for viral spillover events, insofar as we infer more viral host jumps from humans to other animals than from animals to humans. Moreover, we demonstrate heightened evolution in viral lineages that involve putative host jumps. We further observe that the extent of adaptation associated with a host jump is lower for viruses with broader host ranges. Finally, we show that the genomic targets of natural selection associated with host jumps vary across different viral families, with either structural or auxiliary genes being the prime targets of selection. Collectively, our results illuminate some of the evolutionary drivers underlying viral host jumps that may contribute to mitigating viral threats across species boundaries.

The majority of emerging and re-emerging infectious diseases in humans are caused by viruses that have jumped from wild and domestic animal populations into humans (that is, zoonoses)[1]. Zoonotic viruses have caused countless disease outbreaks ranging from isolated cases to pandemics and have taken a major toll on human health throughout history. There is a pressing need to develop better approaches to pre-empt the emergence of viral infectious diseases and mitigate their effects. As such, there is an immense interest in understanding the correlates and mechanisms of zoonotic host jumps[1–10].

Most studies thus far have primarily investigated the ecological and phenotypic risk factors contributing to viral host range through the use of host–virus association databases constructed mainly on the basis of systematic literature reviews and online compendiums, including VIRION[11] and CLOVER[12]. For example, 'generalist' viruses that can infect a broader range of hosts have typically been shown to be

associated with greater zoonotic potential[2,3,5]. In addition, factors such as increasing human population density[1], alterations in human-related land use[4], ability to replicate in the cytoplasm or being vector-borne[3] are positively associated with zoonotic risk. However, despite global efforts to understand how viral infectious diseases emerge as a result of host jumps, our current understanding remains insufficient to effectively predict, prevent and manage imminent and future infectious disease threats. This may partly stem from the lack of integration of genomics into these ecological and phenotypic analyses.

One challenge for predicting viral disease emergence is that only a small fraction of the viral diversity circulating in wild and domestic vertebrates has been characterized so far. Due to resource and logistical constraints, surveillance studies of novel pathogens in animals often have sparse geographical and/or temporal coverage[13,14] and focus on selected host and pathogen taxa. Further, many of these studies do not

[1]UCL Genetics Institute, University College London, London, UK. [2]The Francis Crick Institute, London, UK. [3]These authors contributed equally: Lucy van Dorp, Francois Balloux. ✉e-mail: cedriccstan@gmail.com

perform downstream characterization of the novel viruses recovered and may lack sensitivity due to the use of PCR pre-screening to prioritize samples for sequencing[15]. As such, our knowledge of which viruses can, or are likely to emerge and in which settings, is poor. In addition, while genomic analyses are important for investigating the drivers of viral host jumps[16], most studies do not incorporate genomic data into their analyses. Those that did have mostly focused on measures of host[2] or viral[3] diversity as predictors of zoonotic risk. As such, despite the limited characterization of global viral diversity thus far, existing genomic databases remain a rich, largely untapped resource to better understand the evolutionary processes surrounding viral host jumps.

Further, humans are just one node in a large and complex network of hosts in which viruses are endlessly exchanged, with viral zoonoses representing probably only rare outcomes of this wider ecological network. While research efforts have rightfully focused on zoonoses, viral host jumps between non-human animals remain relatively understudied. Another important process that has received less attention is human-to-animal (that is, anthroponotic) spillover, which may impede biodiversity conservation efforts and could also negatively impact food security. For example, human-sourced metapneumovirus has caused fatal respiratory outbreaks in captive chimpanzees[17]. Anthroponotic events may also lead to the establishment of wild animal reservoirs that may reseed infections in the human population, potentially following the acquisition of animal-specific adaptations that could increase the transmissibility or pathogenicity of a virus in humans[13]. Uncovering the broader evolutionary processes surrounding host jumps across vertebrate species may therefore enhance our ability to pre-empt and mitigate the effects of infectious diseases on both human and animal health.

A major challenge for understanding macroevolutionary processes through large-scale genomic analyses is the traditional reliance on physical and biological properties of viruses to define viral taxa, which is largely a vestige of the pre-genomic era[18]. As a result, taxon names may not always accurately reflect the evolutionary relatedness of viruses, precluding robust comparative analyses involving diverse viral taxa. Notably, the International Committee on Taxonomy of Viruses (ICTV) has been strongly advocating for taxon names to also reflect the evolutionary history of viruses[18,19]. However, the increasing use of metagenomic sequencing technologies has resulted in a large influx of newly discovered viruses that have not yet been incorporated into the ICTV taxonomy. Furthermore, it remains challenging to formally assess genetic relatedness through multiple sequence alignments of thousands of sequences comprising diverse viral taxa, particularly for those that experience a high frequency of recombination or reassortment.

In this study, we leverage the ~12 million viral sequences and associated host metadata hosted on NCBI to assess the current state of global viral genomic surveillance. We additionally analyse ~59,000 viral sequences isolated from various vertebrate hosts using a bespoke approach that is agnostic to viral taxonomy to understand the evolutionary processes surrounding host jumps. We ascertain overall trends in the directionality of viral host jumps between human and non-human vertebrates and quantify the amount of detectable adaptation associated with putative host jumps. Finally, we examine, for a subset of viruses, signatures of adaptive evolution detected in specific categories of viral proteins associated with facilitating or sustaining host jumps. Together, we provide a comprehensive assessment of potential genomic correlates underpinning host jumps in viruses across humans and other non-human vertebrates.

## Results

### An incomplete picture of global vertebrate viral diversity

Global genomic surveillance of viruses from different hosts is key to preparing for emerging and re-emerging infectious diseases in humans and animals[13,16]. To identify the scope of viral genomic data collected thus far, we downloaded the metadata of all viral sequences hosted on NCBI Virus ($n$ = 11,645,803; accessed 22 July 2023; Supplementary Data 1). Most (68%) of these sequences were associated with SARS-CoV-2, reflecting the intense sequencing efforts during the COVID-19 pandemic. In addition, of these sequences, 93.6%, 3.3%, 1.5%, 1.1% and 0.6% were of viruses with single-stranded (ss)RNA, double-stranded (ds)DNA, dsRNA, ssDNA and unspecified genome compositions, respectively. The dominance of ssRNA viruses is not entirely explained by the high number of SARS-CoV-2 genomes, as ssRNA viruses still represent 80% of all viral genomes if SARS-CoV-2 is discounted.

Vertebrate-associated viral sequences represent 93% of this dataset, of which 93% were human associated. The next four most-sequenced viruses are associated with domestic animals (*Sus*, *Gallus*, *Bos* and *Anas*) and, after excluding SARS-CoV-2, represent 15% of vertebrate viral sequences, while viruses isolated from the remaining vertebrate genera occupy a mere 9% (Fig. 1a and Extended Data Fig. 1a), highlighting the human-centric nature of viral genomic surveillance. Further, only a limited number of non-human vertebrate families have at least ten associated viral genome sequences deposited (Fig. 1b), reinforcing the fact that a substantial proportion of viral diversity in vertebrates remains uncharacterized. Viral sequences obtained from non-human vertebrates thus far also display a strong geographic bias, with most samples collected from the United States of America and China, whereas countries in Africa, Central Asia, South America and Eastern Europe are highly underrepresented (Fig. 1c). This geographical bias varies among the four most-sequenced non-human host genera *Sus*, *Gallus*, *Anas* and *Bos* (Extended Data Fig. 1b). Finally, the user-submitted host metadata associated with viral sequences, which is key to understanding global trends in the evolution and spread of viruses in wildlife, remains poor, with 45% and 37% of non-human viral sequences having no associated host information provided at the genus level, or sample collection year, respectively. The proportion of missing metadata also varies extensively between viral families and between countries (Extended Data Fig. 2). Overall, these results highlight the massive gaps in the genomic surveillance of viruses in wildlife globally and the need for more conscientious reporting of sample metadata.

### Humans give more viruses to animals than they do to us

To investigate the relative frequency of anthroponotic and zoonotic host jumps, we retrieved 58,657 quality-controlled viral genomes spanning 32 viral families, associated with 62 vertebrate host orders and representing 24% of all vertebrate viral species on NCBI Virus (https://www.ncbi.nlm.nih.gov/labs/virus/vssi/#/) (Fig. 1d). We found that the user-submitted species identifiers of these viral genomes are poorly ascribed, with only 37% of species names consistent with those in the ICTV viral taxonomy[20]. In addition, the genetic diversity represented by different viral species is highly variable since they are conventionally defined on the basis of the genetic, phenotypic and ecological attributes of viruses[18]. Thus, we implemented a species-agnostic approach based on network theory to define 'viral cliques' that represent discrete taxonomic units with similar degrees of genetic diversity, similar to the concept of operational taxonomic units[21] (Fig. 2a and Methods). A similar approach was previously shown to effectively partition the genomic diversity of plasmids in a biologically relevant manner[22]. Using this approach, we identified 5,128 viral cliques across the 32 viral families that were highly concordant with ICTV-defined species (median adjusted Rand index, ARI = 83%; adjusted mutual information, AMI = 75%) and of which 95% were monophyletic (Fig. 2a). Some clique assignments aggregated multiple viral species identifiers, while others disaggregated species into multiple cliques (Fig. 2b; clique assignments for *Coronaviridae* illustrated in Extended Data Fig. 3). Despite the human-centric nature of genomic surveillance, viral cliques involving only animals represent 62% of all cliques, highlighting the extensive diversity of animal viruses in the global viral-sharing network (Extended Data Fig. 4a).

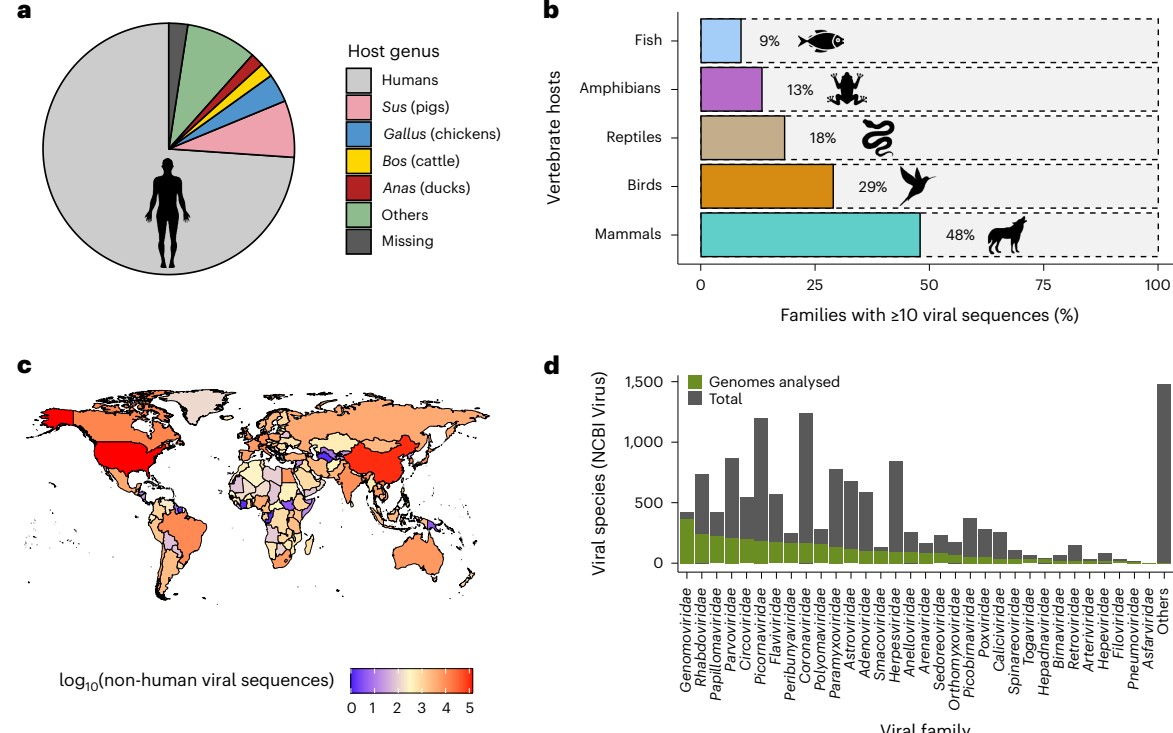

**Fig. 1 | Current state of the global genomic surveillance of vertebrate viruses.**
**a**, Proportion of non-SARS-CoV-2, vertebrate-associated viral sequences deposited in public sequence databases ($n$ = 2,874,732), stratified by host. Viral sequences associated with humans and the next four most-sampled vertebrate hosts are shown. Sequences with no host metadata resolved at the genus level are denoted as 'missing'. **b**, Proportion of host families represented by at least 10 associated viral sequences for the five major vertebrate host groups. **c**, Global heat map of sequencing effort, generated from all viral sequences deposited in public sequence databases that are not associated with human hosts ($n$ = 1,599,672). **d**, Number of vertebrate viral species on NCBI Virus used for the genomic analyses in this study, stratified by viral family. The 32 vertebrate-associated viral families considered in this study are shown and the remaining 21 families that were not considered are denoted as 'others'.

We then identified putative host jumps within these viral cliques by producing curated whole-genome alignments to which we applied maximum-likelihood phylogenetic reconstruction. For segmented viruses, we instead used single-gene alignments as the high frequency of reassortment[23] precludes robust phylogenetic reconstruction using whole genomes. Phylogenetic trees were rooted with suitable outgroups identified using metrics of alignment-free distances (see Methods). We subsequently reconstructed the host states of all ancestral nodes in each tree, allowing us to determine the most probable direction of a host jump for each viral sequence (approach illustrated in Fig. 3a). To minimize the uncertainty in the ancestral reconstructions, we considered only host jumps where the likelihood of the ancestral host state was twofold higher than alternative host states (Fig. 3a and Supplementary Methods). Varying the stringency of this likelihood threshold yielded highly consistent results (Extended Data Fig. 5a), indicating that the inferred host jumps are robust to our choice of threshold. In total, we identified 12,676 viral lineages comprising 2,904 putative vertebrate host jumps across 174 of these viral cliques.

Among the putative host jumps inferred to involve human hosts (599/2,904; 21%), we found a much higher frequency of anthroponotic compared with zoonotic host jumps (64% vs 36%, respectively; Fig. 3b). This finding was statistically significant as assessed via a bootstrap paired $t$-test ($t$ = 227, d.f. = 999, $P$ < 0.0001) and a permutation test ($P$ = 0.035; see Methods). In addition, this result was robust to our choice of likelihood thresholds used during ancestral reconstruction (Extended Data Fig. 5b), the tree depth at which the host jump was identified (Extended Data Fig. 5c), and to sampling bias (Supplementary Notes and Fig. 1). The highest

number of anthroponotic jumps was contributed by the cliques representing SARS-CoV-2 (132/383; 34%), MERS-CoV (39/383; 10%) and influenza A (37/383; 10%). This is concordant with the repeated independent anthroponotic spillovers into farmed, captive and wild animals described for SARS-CoV-2 (refs. 13, 24–27) and influenza A[28,29]. Meanwhile, there has only been circumstantial evidence for human-to-camel transmission of MERS-CoV[30–32]. Noting the disproportionate number of anthroponotic jumps contributed by these viral cliques, we reperformed the analysis without them and found a significantly higher frequency of anthroponotic than zoonotic jumps (53.5% vs 46.5%; bootstrap paired $t$-test, $t$ = 40, d.f. = 999, $P$ < 0.0001), suggesting that our results are not driven solely by these cliques. Further, 16/21 of the viral families were involved in more anthroponotic than zoonotic jumps (Extended Data Fig. 5d), indicating that this finding is generalizable across most viruses. Overall, our results highlight the high but largely underappreciated frequency of anthroponotic jumps among vertebrate viruses.

**Host jumps of multihost viruses require fewer adaptations**
Before jumping to a new host, a virus in its natural reservoir may fortuitously acquire pre-adaptive mutations that facilitate its transition to a new host. This may be followed by the further acquisition of adaptive mutations as the virus adapts to its new host environment[16].

For each host jump inferred, we estimated the extent of both pre-jump and post-jump adaptations through the sum of branch lengths from the observed tip to the ancestral node where the host transition occurred (Fig. 3a). However, in practice, the degree of adaptation inferred may vary on the basis of different factors, including sampling intensity and the time interval between when the host jump

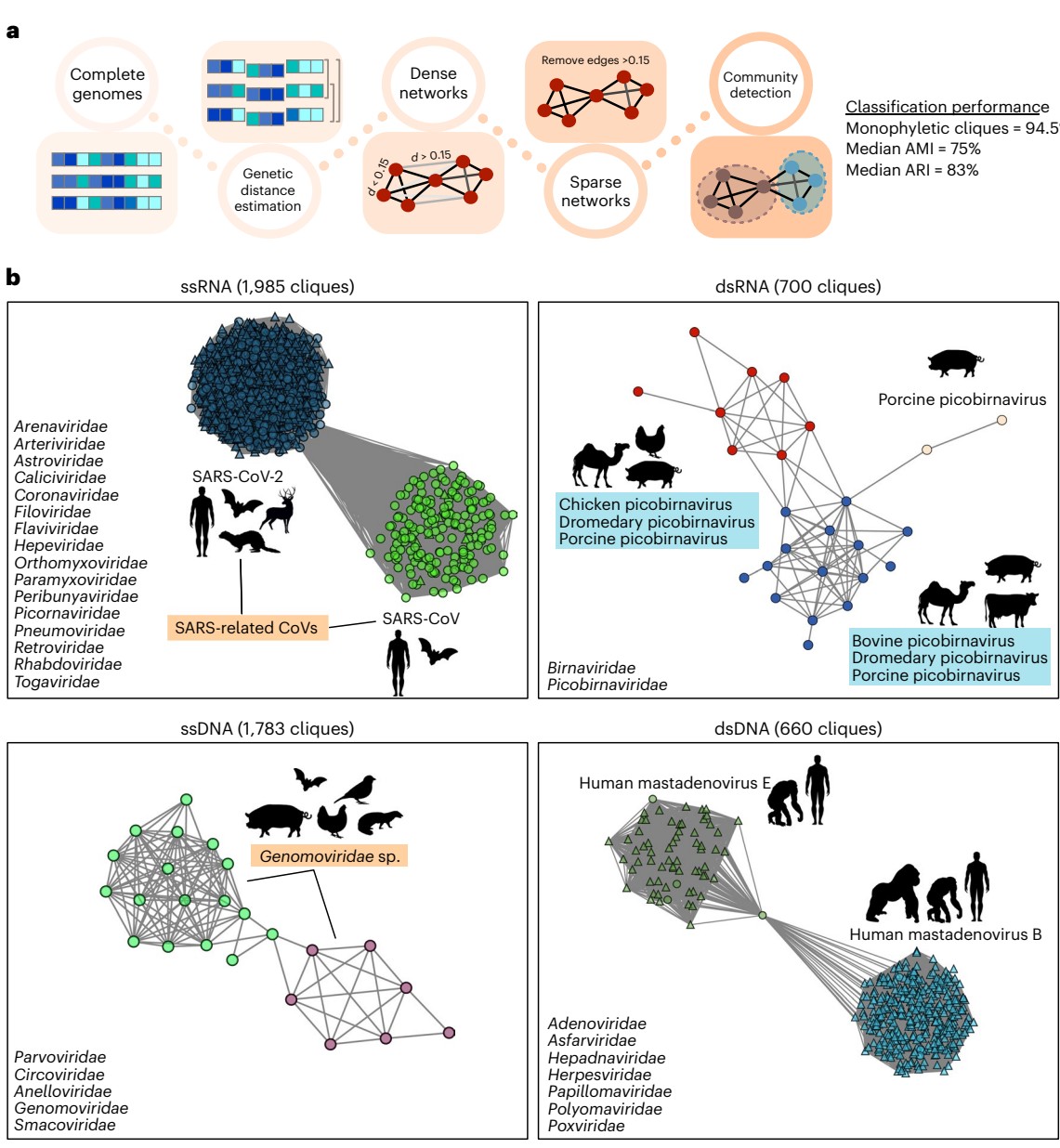

**Fig. 2 | Taxonomy-agnostic approach for identifying equivalent units of viral diversity. a**, Workflow for taxonomy-agnostic clique assignments. Briefly, the alignment-free Mash[53] distances between complete viral genomes in each viral family are computed and dense networks where nodes and edges representing viral genomes and the pairwise Mash distances, respectively, are constructed. From these networks, edges representing Mash distances >0.15 are removed to produce sparse networks, on which the community-detection algorithm, Infomap[54], is applied to identify viral cliques. Concordance with the ICTV taxonomy was assessed using ARI and AMI. **b**, Sparse networks of representative

viral cliques identified within the *Coronaviridae* (ssRNA), *Picobirnaviridae* (dsRNA), *Genomoviridae* (ssDNA) and *Adenoviridae* (dsDNA). Some viral clique assignments aggregated multiple viral species, while others disaggregated species into multiple cliques. Nodes, node shapes and edges represent individual genomes, their associated host and their pairwise Mash distances, respectively. The list of viral families considered in our analysis are shown on the bottom-left corner of each panel. Silhouettes were sourced from Flaticon.com and Adobe Stock Images (https://stock.adobe.com) with a standard licence.

occurred and when the virus was isolated from its new host. As such, for each viral clique, we considered only the minimum mutational distance associated with a host jump.

We first examined whether the minimum mutational distance associated with a host jump for each viral clique was higher than the minimum for a random selection of viral lineages not involved in host jumps (Fig. 3a and Methods). Indeed, the minimum mutational distance for a putative host jump within each clique was significantly higher than that for non-host jumps (Fig. 4a; two-tailed Mann–Whitney $U$-test, $U = 6,767$, $P < 0.0001$). Noting that both sampling intensity and the

different mutation rates of viral families may confound these results, we corrected for these confounders using a logistic regression model but found a similar effect (odds ratio, $OR_{host jump} = 1.31$; two-tailed $Z$-test for slope = 0, $Z = 6.58$, d.f. = 289, $P < 0.0001$).

We then considered the commonly used measure of directional selection acting on genomes, the ratio of non-synonymous mutations per non-synonymous site (dN) to the number of synonymous mutations per synonymous site (dS). Comparing the minimum dN/dS for host jumps within each clique, we observed that minimum dN/dS was also significantly higher for host jumps compared with non-host jumps

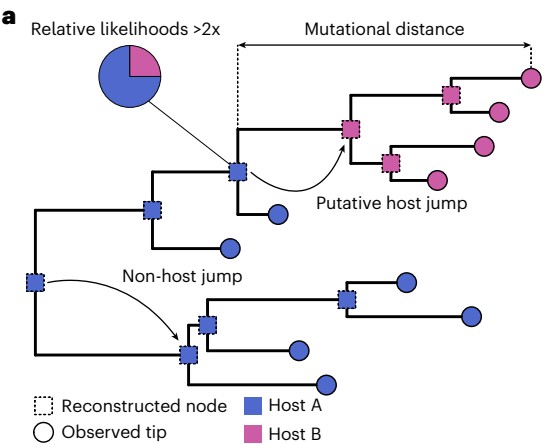

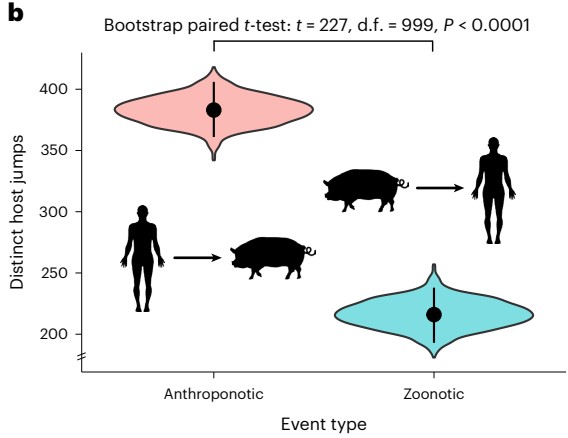

**Fig. 3 | Humans give more viruses to animals than they give to us. a,** Illustration of ancestral host reconstruction approach used to infer the directionality of putative host jumps. Putative host jumps are identified if the ancestral host state has a twofold higher likelihood than alternative host states. The mutational distance (substitutions per site) represents the sum of the branch lengths between the tip sequence and the ancestral node for which the first host state transition occurred in a tip-to-root traverse. **b,** Number of distinct putative host jumps involving humans across all viral families considered ($n = 32$). Black dots represent the observed point estimates for each type of host jump. The violin plots show the bootstrap distributions of these estimates, where the host jumps within each viral clique were resampled with replacement for 1,000 iterations. Black lines show the 95% confidence intervals associated with these bootstrap distributions. Silhouettes were sourced from Flaticon.com and Adobe Stock Images (https://stock.adobe.com) with a standard licence. A two-tailed paired $t$-test was performed to test for a difference in the zoonotic and anthroponotic bootstrap distributions.

(Fig. 4b; $OR_{host jump} = 2.39$; $Z = 4.84$, d.f. = 263, $P < 0.0001$). Finally, after correcting for viral clique membership, there were no significant differences in log-transformed mutational distance ($F_{(1,528)} = 2.23$, $P = 0.136$) or dN/dS estimates ($F_{(1,338)} = 1.66$, $P = 0.198$) between zoonotic and anthroponotic jumps, or between forward and reverse cross-species jumps (mutational distance: $F_{(1,1588)} = 0.538$, $P = 0.463$; dN/dS: $F_{(1,1168)} = 0.0311$, $P = 0.860$), indicating that there are no direction-specific biases in these measures of adaptation. Overall, these results are consistent with the hypothesized heightened selection following a change in host environment and additionally provide confidence in our ancestral-state reconstruction method for assigning host jump status.

However, the extent of adaptive change required for a viral host jump may vary. For instance, some zoonotic viruses may require minimal adaptation to infect new hosts while in other cases, more substantial genetic changes might be necessary for the virus to overcome barriers that prevent efficient infection or transmission in the new host. We therefore tested the hypothesis that the strength of selection associated with a host jump decreases for viruses that tend to have broader host ranges. To do so, we compared the minimum mutational distance between ancestral and observed host states to the number of host genera sampled for each viral clique. We found that the observed host range for each viral clique is positively associated with greater sequencing intensity (that is, the number of viral genomes in each clique; Pearson's $r = 0.486$; two-tailed $t$-test for $r = 0$, $t = 34.9$, d.f. = 3,932, $P < 0.0001$), in line with the strong positive correlation between per-host viral diversity and surveillance effort reported in previous studies[2,3,8]. After correcting for both sequencing effort and viral family membership, we found that the mutational distance for host jumps tends to decrease with broader host ranges (Poisson regression, slope = −0.113; two-tailed $Z$-test for slope = 0, $Z = −9.40$, d.f. = 129, $P < 0.0001$). In contrast, the relationship between mutational distance and host range for viral lineages that have not experienced host jumps is only weakly positive (slope = 0.0843; $Z = 7.16$, d.f. = 127, $P < 0.0001$) (Fig. 4c). Similarly, the minimum dN/dS for a host jump decreases more substantially for viral cliques with broader host ranges (slope = −0.427; $Z = −9.18$, d.f. = 116, $P < 0.0001$) than for non-host jump controls (slope = 0.143; $Z = 3.08$, d.f. = 116, $P < 0.01$) (Fig. 4d). These trends in mutational distance and dN/dS were consistent when the same analysis was performed for ssDNA, dsDNA, +ssRNA and −ssRNA

viruses separately (Extended Data Fig. 6). These results indicate that, on average, 'generalist' multihost viruses experience lower degrees of adaptation when jumping into new vertebrate hosts.

## Host jump adaptations are gene and family specific

We next examined whether genes with different established functions displayed distinctive patterns of adaptive evolution linked to host jump events. Since gene function remains poorly characterized in the large and complex genomes of dsDNA viruses, we focused on the shorter ssRNA and ssDNA viral families. We selected for analysis the four non-segmented viral families with the greatest number of host jump lineages in our dataset: *Coronaviridae* (+ssRNA; $n = 2,537$), *Rhabdoviridae* (−ssRNA; $n = 1,097$), *Paramyxoviridae* (−ssRNA; $n = 787$) and *Circoviridae* (ssDNA; $n = 695$). For these viral families, we extracted all annotated protein-coding regions from their genomes and categorized them as either being associated with cell entry (termed 'entry'), viral replication ('replication-associated') or virion formation ('structural'), and classifying the remaining genes as 'auxiliary' genes.

For the *Coronaviridae*, *Paramyxoviridae* and *Rhabdoviridae*, the entry genes encode surface glycoproteins that could also be considered structural but were not categorized as such given their important role in mediating cell entry. The capsid gene of circoviruses, however, encodes the sole structural protein that is also the key mediator of cell entry and was therefore categorized as structural. To estimate putative signatures of adaptation in relation to lineages that have experienced host jumps for the different gene categories, we modelled the change in $\log_{10}(dN/dS)$ in host jumps versus non-host jumps using a linear model, while correcting for the effects of clique membership (see Methods). Contrary to our expectation that entry genes would generally be under the strongest adaptive pressures during a host jump, we found that the strength of adaptation signals for each gene category varied by family. Indeed, the strongest signals were observed for structural proteins in coronaviruses (effect = 0.375, two-tailed $t$-test for difference in parameter estimates, $t = 4.35$, d.f. = 10,121, $P < 0.0001$) and auxiliary proteins in paramyxoviruses (effect = 0.439, $t = 2.15$, d.f. = 4,225, $P = 0.02$) (Fig. 5). Meanwhile, no significant adaptive signals were observed in the entry genes of all families (minimum $P = 0.3$), except for the capsid gene in circoviruses (effect = 0.325, $t = 2.68$, d.f. = 1,367, $P = 0.004$) (Fig. 5). These findings suggest that selective pressures acting on

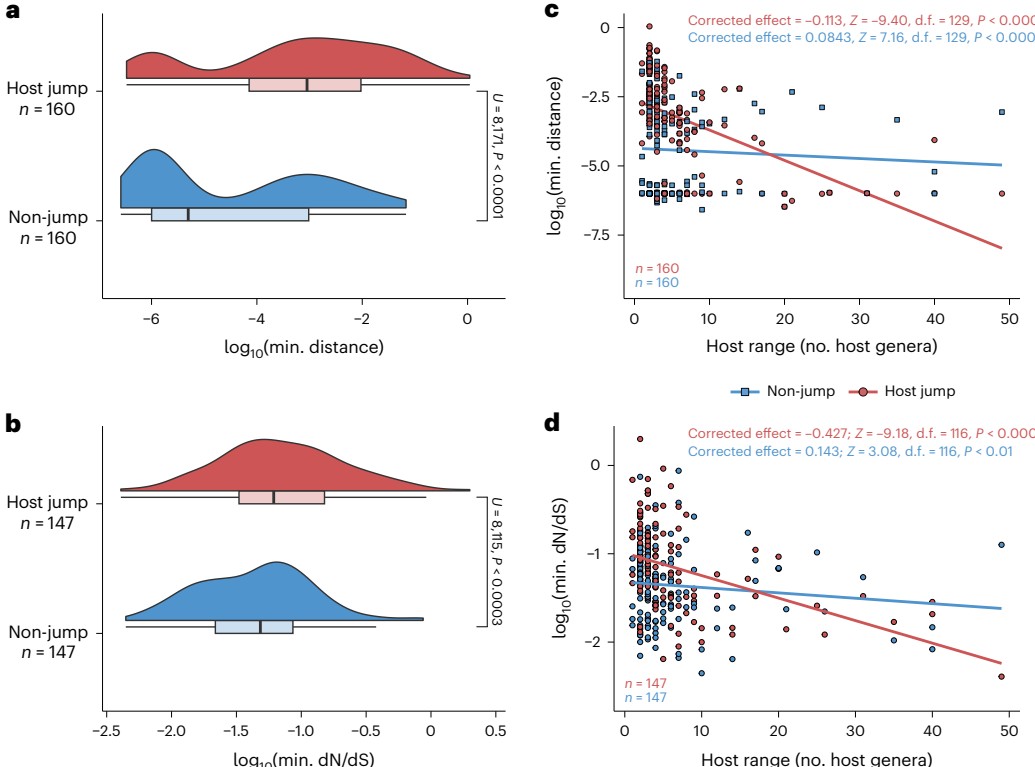

**Fig. 4 | The strength of adaptive signals associated with host jumps decreases with broader viral host ranges. a,b,** Distributions (Gaussian kernel densities and boxplots) of (**a**) minimum mutational distance and (**b**) minimum dN/dS for inferred host jump events and non-host jump controls on the logarithmic scale. Differences in distributions were assessed using two-sided Mann–Whitney *U*-tests. **c,d,** Scatterplots of the (**c**) minimum mutational distance and (**d**) minimum dN/dS for host jump and non-host jumps. Lines represent univariate linear regression smooths fitted on the data. We corrected

for the effects of sequencing effort and viral family membership using Poisson regression models. The parameter estimates in these Poisson models and their statistical significance, as assessed using two-tailed *Z*-tests, after performing these corrections are annotated. For all panels, each data point represents the minimum distance or minimum dN/dS across all host jump or randomly selected non-host jump lineages in a single clique. Boxplot elements are defined as follows: centre line, median; box limits, upper and lower quartiles; whiskers, 1.5× interquartile range.

viral genomes in relation to host jumps are likely to differ by gene function and viral family.

Given the lack of adaptive signals in the entry proteins, we further hypothesized that within each gene, adaptive changes are likely to be localized to regions of functional importance and/or that are under relatively stronger selective pressures exerted by host immunity. To test this, we focused on the spike gene (entry) of viral cliques within the *Coronaviridae* since the key region involved in viral entry is well characterized (that is, the receptor-binding domain (RBD))[33]. We found that dN/dS estimates consistent with adaptive evolution were indeed localized to the RBDs, but also to the N-terminal domains (NTD), of SARS-CoV-2 (genus *Betacoronavirus*), avian infectious bronchitis virus (IBV; *Gammacoronavirus*) and MERS (genus *Alphacoronavirus*) (Extended Data Fig. 7). This is consistent with the strong immune pressures exerted on these regions of the spike protein[34,35] and the central role of the RBD in host-cell recognition and entry[36–38]. Overall, our results indicate that the extent of adaptation associated with a host jump likely varies by gene function, gene region and viral family.

## Discussion

The post-genomic era has opened opportunities to advance our understanding of the diversity of viruses in circulation and the macroevolutionary principles of viral host range. Leveraging ~59,000 publicly available viral sequences isolated from vertebrate hosts, we inferred that humans give more viruses to other vertebrates than they give to us across the 32 viral families we considered. We further demonstrated that host jumps are associated with heightened signals of adaptive evolution that tend to decrease in viruses with broader host ranges.

This indicates that there may be a minimum mutational threshold necessary for viruses to expand their host range. Finally, we showed that adaptive evolution linked to host jumps may vary by gene function and may be localized to specific gene regions of functional importance.

To bypass the limitations of existing viral taxonomies, we used a taxonomy-agnostic approach to define roughly equivalent units of viral diversity, which formed the basis for most of the analyses presented in this study. The use of operational taxonomic units rather than traditional taxonomic species names further allowed us to perform like-for-like analyses across the entire diversity of viruses. Our approach identified cliques that were largely concordant with traditional viral species nomenclature but also highlighted inconsistencies, where in some cases, single viral species appear to form distinct taxonomic groups while other groups of species seem to form a single group solely based on their genetic relatedness (Fig. 2 and Extended Data Fig. 3). However, we do not claim that our approach should supersede existing taxonomic classification systems, especially since a robust and meaningful species definition requires the integration of viral properties with finer-scale evolutionary analyses that was not necessary for our purposes. Nevertheless, we anticipate that the development and use of similar network-based approaches may pave the way for the development of efficient classification frameworks that can rapidly incorporate novel, metagenomically derived viruses into existing taxonomies.

Harnessing cliques as a mechanism of identifying clusters of related viruses for phylogenetic inspection allowed us to quantify the number and sources of recent host jump events. One important caveat to this approach is that the viral cliques involved in putative host jumps represent only a fraction of the viral diversity sequenced thus

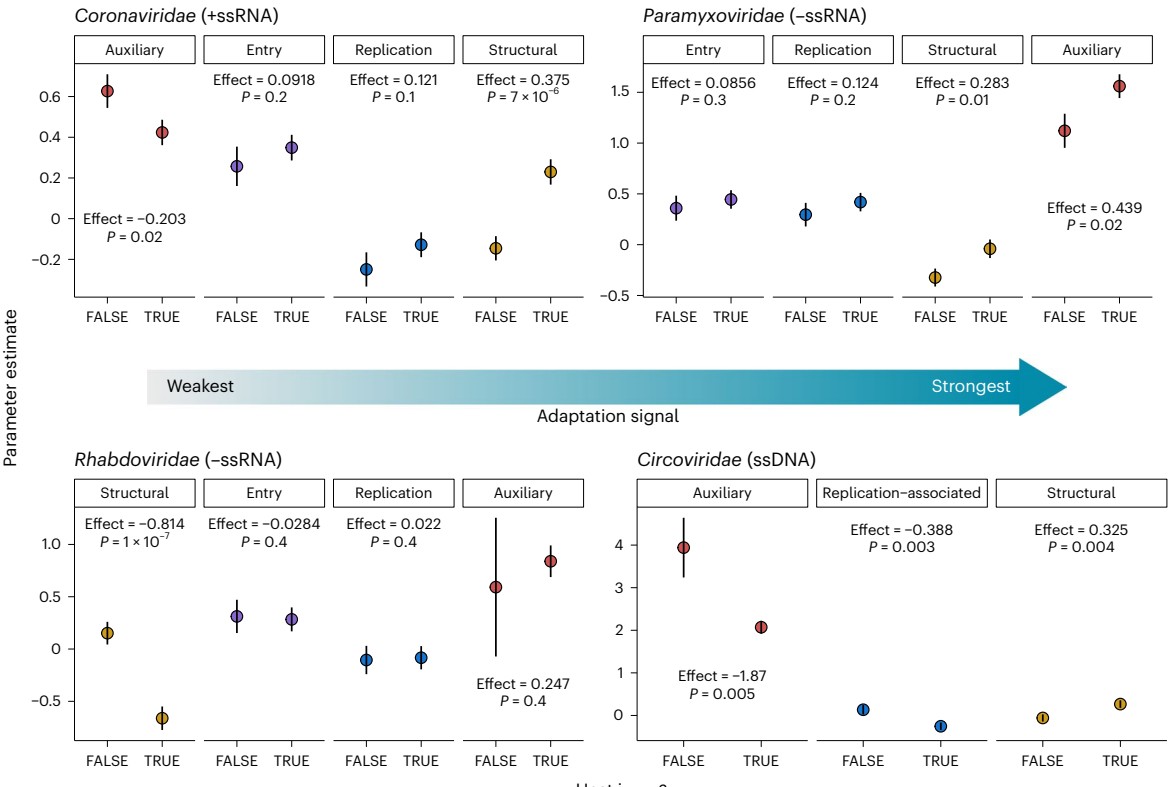

**Fig. 5 | Signals of adaptation are gene and family specific.** The strength of adaptation signals in genes associated with host jump and non-host jump lineages were estimated using linear models for *Coronaviridae* (*n* = 10,129), *Paramyxoviridae* (*n* = 4,233), *Rhabdoviridae* (*n* = 3,321), and *Circoviridae* (*n* = 1,373). We modelled the effects of gene type and host jump status on log(dN/dS) while correcting for viral clique membership and, for each gene type, inferred the strength of adaptive signal (denoted 'effect') as the difference in parameter estimates for host jumps versus non-host jumps. Points and lines represent the parameter estimates and their standard errors, respectively. Differences in parameter estimates were tested against zero using a one-tailed *t*-test. Subpanels for each gene type were ordered from left to right with increasing effect estimates.

far (Extended Data Fig. 4b) and the patterns we observed may change as more viruses are discovered. However, we consistently found higher frequencies of anthroponotic than zoonotic jumps across 16 of the 21 viral families (Extended Data Fig. 5d). Since each of these families are associated with varying viral discovery effort, the consistency of this pattern makes it highly unlikely that surveillance biases are driving the excess of anthroponotic jumps we inferred. Another caveat is that our clique assignment approach clusters viruses within ~15% sequence divergence, which limits our analyses to relatively recent host jump events. However, the limited divergence of the sequences within each clique also allowed us to produce more robust alignments and hence evolutionary inferences.

Of the 599 recent host jumps identified, 64% were inferred as anthroponotic (Fig. 3b). While the relative importance of anthroponotic versus zoonotic events has been speculated[13,29,39,40], we provide a formal evaluation of the zoonotic-to-anthroponotic ratio in vertebrates, showing that anthroponoses are equally, if not more, critical to consider than zoonoses when assessing viral spillover dynamics. It stands to reason that the substantial global human population size and ubiquitous spatial distribution position us as a major source for viral exchange. However, it is also likely that behavioural factors might amplify the risk of anthroponotic transmission, for example, through changes in land use, agricultural methods or heightened interactions between humans and wildlife[4]. Overall, our results highlight the importance of surveying and monitoring human-to-animal transmission of viruses, and its impacts on human and animal health.

We observed heightened evolution and adaptive signals in association with host jumps (Fig. 4). This result is largely intuitive, since a virus jumping into a new host is likely to be under different selective

pressures exerted directly by the novel host environment and indirectly by changes in host-to-host transmission dynamics. The evolutionary signals we captured may include pre-requisite adaptations that enable a virus to infect the new host. In addition, they probably also represent the burst of adaptive mutations which may be acquired following a host jump, which has been demonstrated for multiple viral systems[24,41–43]. Further, these signals could potentially reflect a relaxation of previous selective pressures no longer present in the novel host. We note that these signals of heightened evolution could also, in principle, be inflated by sampling bias, where two viruses circulating in the same host are more often drawn from the same population. However, this was largely controlled for in our analysis through comparisons to representative non-host jump lineages that are expected to be affected by the same sampling bias.

We observed lower mutational and adaptive signals associated with host jumps for viruses that infect a broader range of hosts (Fig. 4c,d). The most likely explanation for this pattern is that some viruses are intrinsically more capable of infecting a diverse range of hosts, possibly by exploiting host-cell machinery that are conserved across different hosts. For example, sarbecoviruses (the subgenus comprising SARS-CoV-2) target the ACE2 host-cell receptor, which is conserved across vertebrates[44,45], and the high structural conservation of the sarbecovirus spike protein[15] may explain the observation that single mutations can enable sarbecoviruses to expand their host tropism[46]. In other words, multihost viruses may have evolved to target more conserved host machinery that reduces the mutational barrier for them to productively infect new hosts. This may provide a mechanistic explanation for previous observations that viruses with broad host range have a higher risk of emerging as zoonotic diseases[2,3,5].

Our approach to identifying putative host jumps hinges on ancestral-state reconstruction (Fig. 3a), which has been shown to be affected by sampling biases[47,48]. However, we accounted for this, at least in part, by including sequencing effort as a measure of sampling bias in our statistical models, allowing us to draw inferences that were robust to disproportionate sampling of viruses in different hosts. Our approach also does not consider the epidemiology or ecology of viral transmission, as this is largely dependent on host features such as population size, social structure and behaviour for which comprehensive datasets at this scale are not currently available. We anticipate that future datasets that integrate ecology, epidemiology and genomics may allow more granular investigations of these patterns in specific host and viral systems. In addition, the patterns we described are broad and do not capture the idiosyncrasies of individual host–pathogen associations. These include a variety of biological features — intrinsic ones, such as the molecular adaptations required for receptor binding, as well as more complex ones including cross-immunity and interference with other viral pathogens circulating in a host population.

Overall, our work highlights the large scope of genomic data in the public domain and its utility in exploring the evolutionary mechanisms of viral host jumps. However, the large gaps in the genomic surveillance of viruses thus far suggest that we have only just scratched the surface of the true viral diversity in nature. In addition, despite the strong anthropocentric bias in viral surveillance, 81% of the putative host jumps identified in this study do not involve humans, emphasizing the large underappreciated scale of the global viral-sharing network (Extended Data Fig. 8). Widening our field of view beyond zoonoses and investigating the flow of viruses within this larger network could yield valuable insights that may help us better prepare for and manage infectious disease emergence at the human–animal interface.

## Methods

### Data acquisition, curation and quality control

The metadata of all partial and complete viral genomes were downloaded from NCBI Virus (https://www.ncbi.nlm.nih.gov/labs/virus/vssi/#/) on 22 July 2023, with filters excluding sequences isolated from environmental sources, lab hosts, or associated with vaccine strains or proviruses ($n$ = 11,645,803). Where possible, host taxa names in the metadata were resolved in accordance with the NCBI taxonomy[49] using the 'taxizedb' v.0.3.1 package in R. User-submitted viral species names were compared to the ICTV master species list version 'MSL38. V2' dated 6 July 2023.

To generate a candidate list of viral sequences for further genomic analysis, the metadata were filtered to include 53 viral families known to infect vertebrate hosts on the basis of information provided in the 2022 release of the ICTV taxonomy (https://ictv.global/taxonomy)[50] and with reference to that provided by ViralZone (https://viralzone.expasy.org/)[51]. We then retained only sequences from viral families comprising at least 100 sequences of greater than 1,000 nt in length. Since the sequences of segmented viral families are rarely deposited as whole genomes and since the high frequency of reassortment[23] precludes robust phylogenetic reconstruction, we identified sequences for single genes conserved within each of these families for further analysis (*Arenaviridae*: L segment; *Birnaviridae*: ORF1/RdRP/VP1/Segment B; *Peribunyaviridae*: L segment; *Orthomyxoviridae*: PB1; *Picobirnaviridae*: RdRP; *Sedoreoviridae*: VP1/Segment 1/RdRP; *Spinareoviridae*: Segment 1/RdRP/Lambda 3). These sequences were retrieved by applying text-based pattern matching (that is, 'grepl' in R) to query the GenBank sequence titles. For non-segmented genomes, we retained all non-human-associated sequences and subsampled the human-associated sequences as follows: we selected a random subsample of 1,000 SARS-CoV-2 genomes of greater than 28,000 nt from distinct countries, isolation sources and with distinct collection dates. For influenza B, we retained only human sequences with distinct country of origins, sample types and collection dates, and

hosts of isolation. For other human-associated sequences, we retained viruses with distinct species, country, isolation source and collection date information. We then downloaded the final candidate list of viral sequences ($n$ = 92,973) using 'ncbi-acc-download' v.0.2.8 (https://github.com/kblin/ncbi-acc-download). Further quality control of the genomes downloaded was performed using 'CheckV' (v.1.0.1)[52], retaining sequences with more than 95% completeness (for non-segmented viruses) and less than 5% contamination (for all sequences). This resulted in a final genomic dataset comprising 58,657 observations (Supplementary Table 1) composed of gene sequences for segmented viruses and complete genomes for non-segmented viruses. For simplicity, we will henceforth refer to the gene sequences and complete genomes as 'genomes'.

### Taxonomy-agnostic identification of viral cliques

To identify viral cliques, we calculated the pairwise alignment-free Mash distances of genomes within each viral family via 'Mash' (v.1.1)[53] with a $k$-mer size of 13. This $k$-mer size ensures that the probability of observing a $k$-mer by chance, given the median genome length for each clique, is less than 0.01. Given a genome length, $l$, alphabet, $\Sigma$ = {A, T, G, C}, and the desired probability of observing a $k$-mer by chance, $q$ = 0.01, this was computed using the formula described previously[53]:

$$k = \left\lceil \log_{|\Sigma|}\left(\frac{l(1-q)}{q}\right) \right\rceil \tag{1}$$

We then constructed undirected graphs for each viral family with nodes and edges representing genomes and Mash distances, respectively. From these networks, we removed edges with Mash distance values greater than a certain threshold, $t$, before we applied the community-detection algorithm, Infomap[54]. This community-detection algorithm performs well in both large (>1,000 nodes) and small (≤1,000 nodes) undirected graphs[55] and seeks to identify subgraphs within these undirected graphs that minimize the information required to constrain the movement of a random walker[54]. We refer to the subgraphs identified through this algorithm as 'viral cliques'. Here we forced the community-detection algorithm to identify taxonomically relevant cliques by removing edges with Mash distance values greater than $t$, which resulted in sparser graphs with closely related genomes (for example, from the same species) being more densely connected than more distantly related genomes (for example, different species). The value of $t$ was selected by maximizing the proportion of monophyletic cliques identified and the concordance of the viral cliques identified with the viral species names from the NCBI taxonomy, based on the commonly used clustering performance metrics, AMI and ARI (Supplementary Fig. 2). These metrics were computed using the 'AMI' and 'ARI' functions in 'Aricode' v.1.0.2. To assess whether the viral cliques identified fulfil the species definition criterion of being monophyletic[18], we reconstructed the phylogenies of each viral family by applying the neighbour-joining algorithm[56] implemented in the 'Ape' v.5.7.1 R package on their pairwise Mash distance matrices. We then computed the proportion of monophyletic viral cliques using the 'is.monophyletic' function in Ape v.5.7.1 across the various values of $t$. Given the discordance between the NCBI and ICTV taxonomies, we applied the above optimization protocol to $t$ using the viral species names in the ICTV taxonomy. Using the NCBI viral species names, $t$ = 0.15 maximized both the median AMI and ARI across all families (Supplementary Fig. 2a), with 94.3% of the cliques identified being monophyletic (Supplementary Fig. 2b). Using the ICTV viral species names, $t$ = 0.2 and $t$ = 0.25 maximized the median AMI and median ARI across families (Supplementary Fig. 2c), with 93.7% and 87.8% of the cliques being monophyletic (Supplementary Fig. 2b), respectively. Since $t$ = 0.15 produced the highest proportion of monophyletic clades that were approximately concordant with existing viral taxonomies, we used this threshold to generate the final viral clique assignments for downstream analyses (Supplementary Table 1).

## Identification of putative host jumps

We retrieved all viral cliques that were associated with at least two distinct host genera and comprised at least 10 genomes ($n$ = 215). We then generated clique-level genome alignments using the 'FFT-NS-2' algorithm in 'MAFFT' (v.7.490)[57,58]. We masked regions of the alignments that were poorly aligned or prone to sequencing error by replacing alignment sites that had more than 10% of gaps or ambiguous nucleotides with Ns. Clique-level genome alignments that had more than 20% of the median genome length masked were considered to be poorly aligned and thus removed from further analysis ($n$ = 6; Supplementary Fig. 3). Following this procedure, we reconstructed maximum-likelihood phylogenies for each viral clique with 'IQ-Tree' (v.2.1.4-beta)[59], using 1,000 ultrafast bootstrap (UFBoot)[60] replicates. The optimal substitution model for each tree was automatically determined using the 'ModelFinder'[61] utility native to IQ-Tree. To estimate the root position for each clique tree, we reconstructed neighbour-joining Mash trees for each viral clique, including 10 additional genomes whose minimum pairwise Mash distance to the genomes in each tree was 0.3–0.5, as potential outgroups. The most basal tips in these neighbour-joining Mash trees were identified and used to root the maximum-likelihood clique trees. This approach, as opposed to using maximum-likelihood phylogenetic reconstruction involving the outgroups, was used as it is difficult to reliably align clique sequences with highly divergent outgroups.

To identify putative host jumps, we performed ancestral-state reconstruction on the resultant rooted maximum-likelihood phylogenies with host as a discrete trait using the 'ace' function in Ape v.5.7.1. Traversing from a tip to the root node, a putative host jump is identified if the reconstructed host state of an ancestral node is different from the observed tip state, has a twofold greater likelihood compared with alternative states and is different from the host state of the sampled tip. Where the tip and ancestral host states were of different taxonomic ranks, we excluded putative host jumps where the ancestral host state is nested within the tip host state, or vice versa (for example, '*Homo*' and '*Hominidae*'). Missing host metadata were encoded as 'unknown' and included in the ancestral-state reconstruction analysis. Host jumps involving unknown or non-vertebrate host states were excluded from further analysis. Separately, we extracted non-host jump lineages to control for any biases in our analysis approach. To do so, we randomly selected an ancestral node where the reconstructed host state is the same as the observed tip state and has a twofold greater likelihood than alternative host states, for each viral genome that is not involved in any putative host jumps. For the mutational distance and dN/dS analyses, we retained only viral cliques where non-host jump lineages could be identified. An analysis exploring the robustness of this host jump inference approach to sampling biases (Supplementary Fig. 1) and a more detailed description of the inference algorithm (Supplementary Fig. 4) are provided in Supplementary Information.

Implementation of this algorithm yielded a list of all viral lineages involving a host jump (Supplementary Table 2). Since multiple lineages may involve a host transition at the same ancestral node, we calculated the number of unique host jump events as the number of distinct nodes for each unique host pair. For example, the three lineages Node1 (host A)→Tip1 (host B), Node1 (host A)→Tip2 (host B) and Node1 (host A)→Tip3 (host C) would be considered as two distinct host jump events, one between hosts A and B and the other between hosts A and C. This counting approach was used for Fig. 3a and Extended Data Fig. 5. The list of all 2,904 distinct host jumps is provided in Supplementary Table 3.

## Calculating mutational distances and dN/dS

Mutational distance and dN/dS estimates may be lineage specific and may depend on sampling intensity. In addition, there is a nonlinear relationship between dN/dS and branch length, that is, the estimated dN/dS decreases with increasing evolutionary distance[62]. Therefore, we opted to compare the minimum adaptive signal (that is, minimum dN/dS) associated with a host jump for each clique. For host jump lineages, mutational distances were calculated as the sum of the branch lengths between the tip sequence and the ancestral node for which the first host state transition occurred (in substitutions per site) using the 'get_pairwise_distances' function in the 'Castor' (v.1.7.10)[63] R package; this was then multiplied by the alignment length to obtain the estimated number of substitutions (Fig. 3a). To calculate the dN/dS estimates, we reconstructed the ancestral sequences of ancestral nodes using the '-asr' flag in IQ-Tree, which is based on an empirical Bayesian algorithm (http://www.iqtree.org/doc/Command-Reference). We then extracted coding regions from the clique-level masked alignments based on the user-submitted gene annotations on NCBI GenBank (in 'gff' format) of each viral genome. We then computed the dN/dS estimates using the method of ref. 64 implemented in the 'dnastring2kaks' function of the 'MSA2dist' v.1.4.0 R package (https://github.com/kullrich/MSA2dist). We calculated the minimum mutational distance and dN/dS across all host jump events in each clique for our downstream statistical analyses, which, in principle, represents the minimum evolutionary signal associated with a host jump in each viral clique. For non-host jump lineages, we similarly computed the minimum mutational distance and dN/dS across the randomly selected lineages. Estimates where dN = 0 or dS = 0 were removed. The list of all minimum mutational distance and minimum dN/dS estimates is provided in Supplementary Tables 4 and 5, respectively. The dN/dS estimates for the analysis shown in Fig. 5 are provided in Supplementary Table 6.

For the coronavirus spike gene analysis (Extended Data Fig. 7), spike sequences were extracted from the clique-level multiple sequence alignments, with gaps trimmed to the reference sequences (avian infectious bronchitis virus, EU714028.1; SARS-CoV-2, MN908947.3; MERS, JX869059.2). The genomic coordinates for the functional domains of the spike proteins were derived from previous studies[33,37,65]. Estimates where dN = 0 or dS = 0 were removed. The dN/dS estimates are provided in Supplementary Table 7.

## Statistical analyses

All statistical analyses were performed using the 'stats' package native to R v.4.3.1. To generate the bootstrapped distributions shown in Fig. 3b, we randomly resampled the host jumps within each clique with replacement (1,000 iterations) and performed two-tailed paired $t$-tests using the 't.test' function. Mann–Whitney $U$-tests, analysis of variance (ANOVA), linear regressions, and Poisson and logistic regressions were implemented using 'wilcox.test', 'anova', 'lm' and 'glm' functions, respectively.

A permutation test was performed to assess whether the higher proportion of anthroponotic versus zoonotic jumps was statistically significant. We randomly permuted the host states in each clique for 500 iterations while preserving the number of host-jump and non-host-jump lineages (illustrated in Supplementary Fig. 5). The $P$ value was calculated as the number of iterations where the permutated anthroponotic/zoonotic ratio was greater than or equal to the observed ratio.

To assess the relationship between host range and adaptive signals (Fig. 4), we used Poisson regressions to model the expected number of host genera observed in each viral clique, $\lambda_{\text{host range}}$. We corrected for the number of genomes in each clique, $g$, as a measure of sampling effort, and viral family membership, $v$, by including them as fixed effects in these models. These models can be formalized for mutational distance or dN/dS, $d$, with some $p$ number of viral families and residual error, $\varepsilon$, as:

$$\ln(\lambda_{\text{hostrange}}) = \beta_0 + \beta_1(\ln(g)) + \sum_{i=1}^{p} \beta_{i+1} v_i + \beta_{p+2}(\ln(d)) + \varepsilon \quad (2)$$

We tested whether the parameter estimates were non-zero by performing two-tailed $Z$-tests implemented within the 'summary' function in R.

To estimate the strength of adaptive signals for coronaviruses, paramyxoviruses, rhabdoviruses and circoviruses (Fig. 5) by gene type, we implemented two linear regression models for each viral family. Since the overall adaptive signal may differ for each viral clique, we corrected for this effect by using an initial linear model where the number of viral cliques, viral clique membership and residual are given by $q$, $c$ and $\varepsilon$, respectively, as follows:

$$\ln(dN/dS) = \beta_0 + \sum_{i=1}^{q} \beta_i c_i + \beta_{p+2}(\ln(d)) + \varepsilon + \varepsilon_{\text{model1}} \quad (3)$$

Subsequently, we used the corrected log(dN/dS) estimates represented by the residuals of model 1, $\varepsilon_{\text{model1}}$, in a second linear model partitioning the effects of gene type by host jump status, $j$. Given $r$ number of gene types, this model can be formalized as follows:

$$\varepsilon_{\text{model1}} = \beta_0 + \sum_{i=1}^{r} \sum_{j=1}^{2} \beta_{i,j} c_{i,j} + \varepsilon_{\text{model2}} \quad (4)$$

The estimated effects shown in Fig. 5, representative of the difference in adaptive signals associated with jump and non-host jump lineages for each gene type, were then computed as:

$$\text{Effect} = \beta_{r,\text{jump}} - \beta_{r,\text{non-jump}} \quad (5)$$

To test whether this effect is statistically significant, we used a one-tailed $t$-test, with the $t$ statistic computed using the standard error of the parameter estimates in model 2:

$$t_r = \frac{\text{Effect}}{\sqrt{\text{s.e.}_{\beta_{r,\text{jump}}}^2 + \text{s.e.}_{\beta_{r,\text{non-jump}}}^2}} \quad (6)$$

The residuals of model 2 were confirmed to be approximately normal by visual inspection (Supplementary Fig. 6).

### Data analysis and visualization
All data analyses were performed using R v.4.3.1. All visualizations were performed using ggplot (v.3.4.2)[66] or ggtree (v.3.8.2)[67]. UpSet plots were created using the R package, UpSetR (v.1.4.0)[68].

### Reporting summary
Further information on research design is available in the Nature Portfolio Reporting Summary linked to this article.

## Data availability
The full list of accessions considered in this study is provided in Supplementary Data 1. The data used for the main analyses are provided in Supplementary Tables 2–7. All reconstructed maximum-likelihood trees and ancestral sequences used for the analyses are hosted on Zenodo (https://doi.org/10.5281/zenodo.10214868)[69].

## Code availability
All custom code used to perform the analyses reported here are hosted on GitHub (https://github.com/cednotsed/vertebrate_host_jumps).

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

## Acknowledgements

We thank R. J. Gibbs, G. Murray and L. P. Shaw for helpful feedback and discussions. C.C.S.T. was funded by the National Science Scholarship

from the Agency for Science, Technology and Research (A*STAR), Singapore. F.B. and L.v.D. were funded by the European Commission (Horizon 2021–2024, END-VOC Project). L.v.D. was also funded by the UCL Excellence Fellowship. Views and opinions expressed are, however, those of the authors only and do not necessarily reflect those of the European Union or the European Health and Digital Executive Agency. For the purpose of open access, the corresponding author has applied a 'Creative Commons Attribution' (CC BY) licence to any author-accepted version of the manuscript. The authors acknowledge the use of the UCL Myriad High Performance Computing Facility (Myriad@UCL), the UCL Department of Computer Science High Performance Computing Cluster and associated support services in the completion of this work.

## Author contributions

C.C.S.T. performed all analyses. L.v.D. and F.B. jointly supervised the study. C.C.S.T., L.v.D. and F.B. wrote the manuscript.

## Competing interests

The authors declare no competing interests.

## Additional information

**Extended data** is available for this paper at https://doi.org/10.1038/s41559-024-02353-4.

**Correspondence and requests for materials** should be addressed to Cedric C. S. Tan.

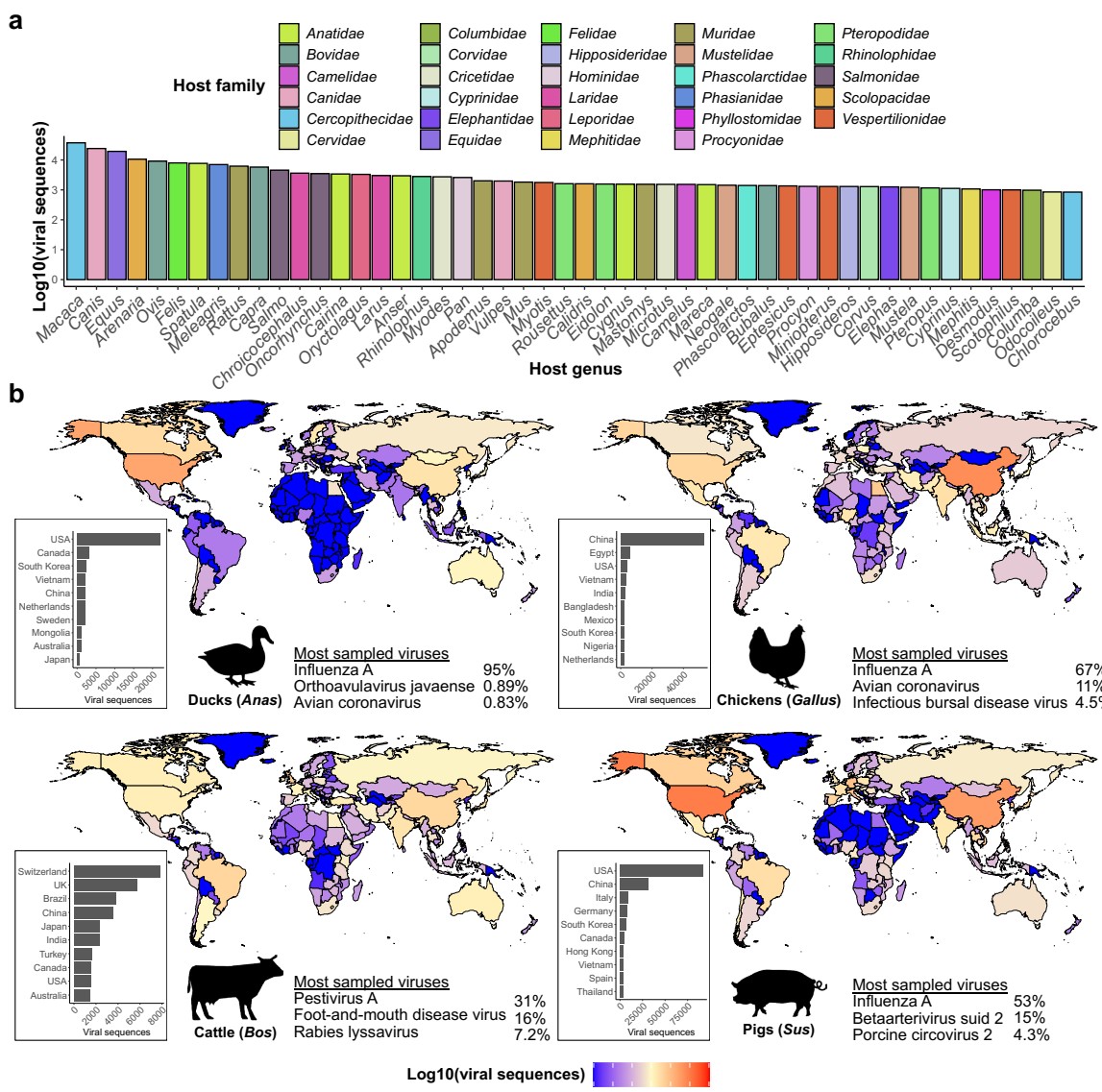

**Extended Data Fig. 1 | Host and geographical distribution of viral sequences.**
(**a**) Number of viral sequences, excluding SARS-CoV-2, associated with the top 50 vertebrate hosts observed in the 'others' category as shown in main text Fig. 1a. (**b**) Number of viral sequences stratified by the four most-sequenced non-human animals, excluding SARS-CoV-2. The number of viral sequences for the top 10 countries are shown as bar plots. The percentage of viral sequences for the top three most sequenced viral species for each host are annotated.

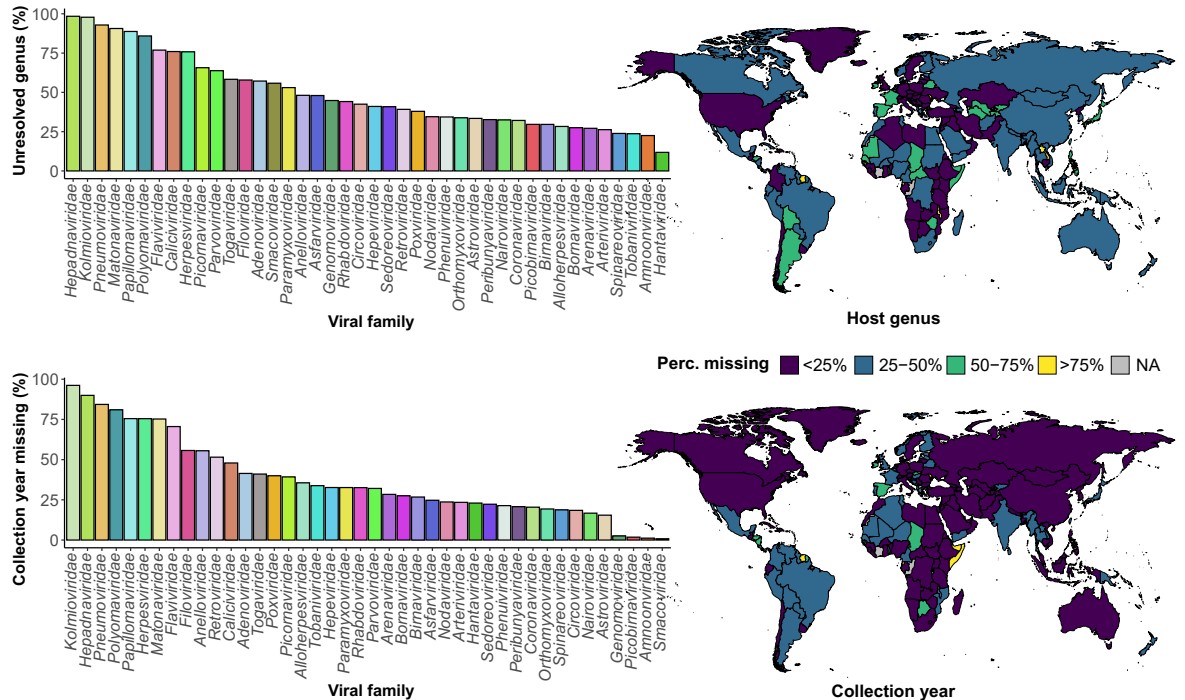

**Extended Data Fig. 2 | Distribution of missing metadata for viral sequences.** (Top) Proportion of all viral sequences associated to non-human vertebrates (*n* = 1,599,672) with missing genus information or (bottom) sample collection year, stratified by viral family or country of origin. Countries with no associated sequences are denoted 'NA'.

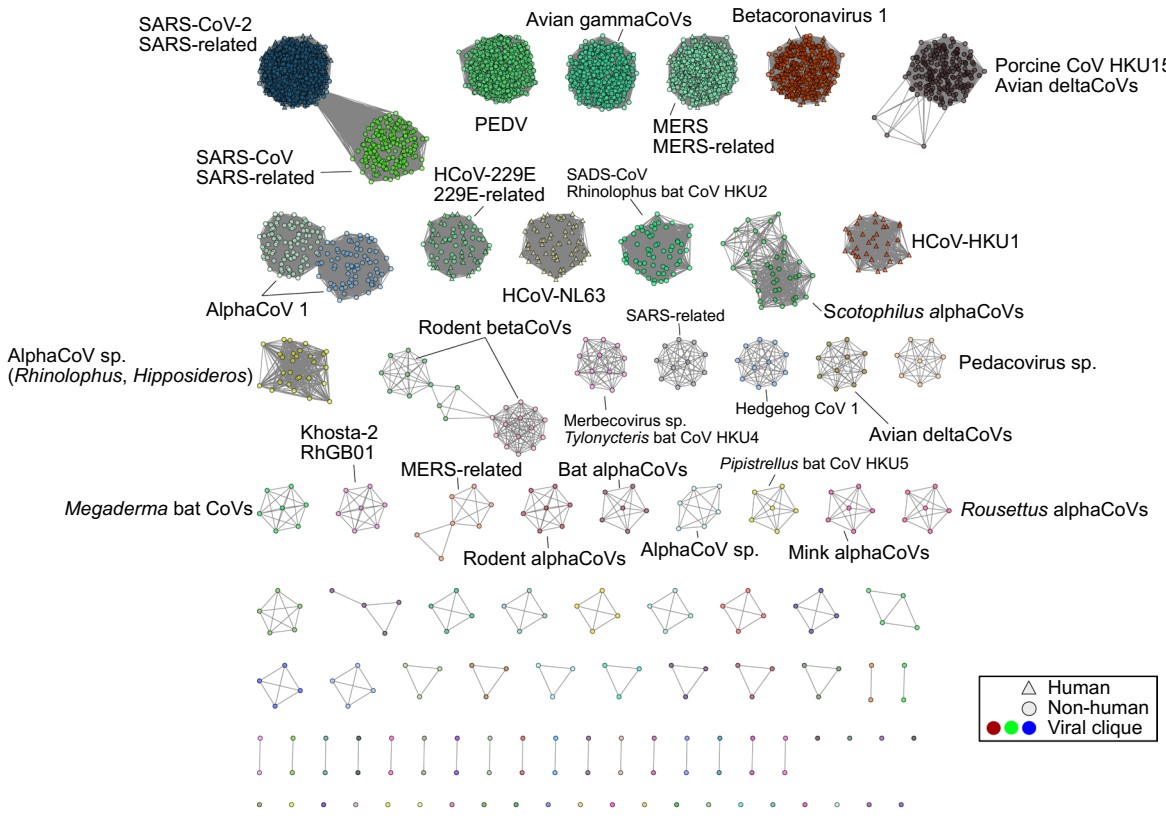

**Extended Data Fig. 3 | Viral cliques for *Coronaviridae*.** Sparse networks of viral cliques identified (see Methods) and their corresponding user-submitted species names for the *Coronaviridae*, similar to main text Fig. 2. Nodes, node shapes, and edges represent individual genomes, their associated host and their pairwise Mash (alignment-free) distances, respectively.

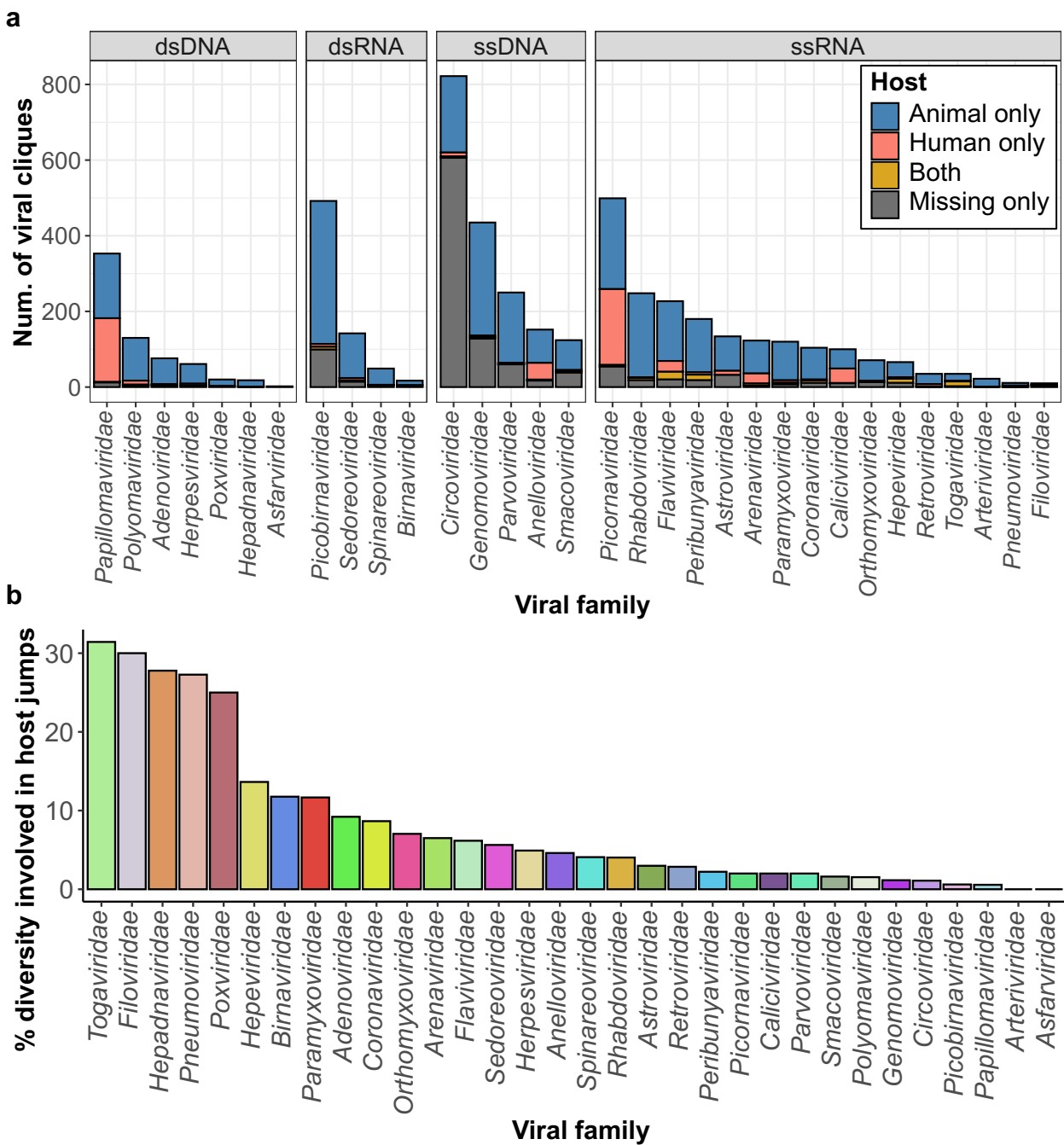

**Extended Data Fig. 4 | Summary of viral cliques identified. (a)** Number of viral cliques identified stratified by viral family. Cliques with only animal-associated sequences, human-associated sequences, or both are annotated. **(b)** Percentage of viral cliques involving at least one of the 2,904 putative host jumps inferred, stratified by viral family.

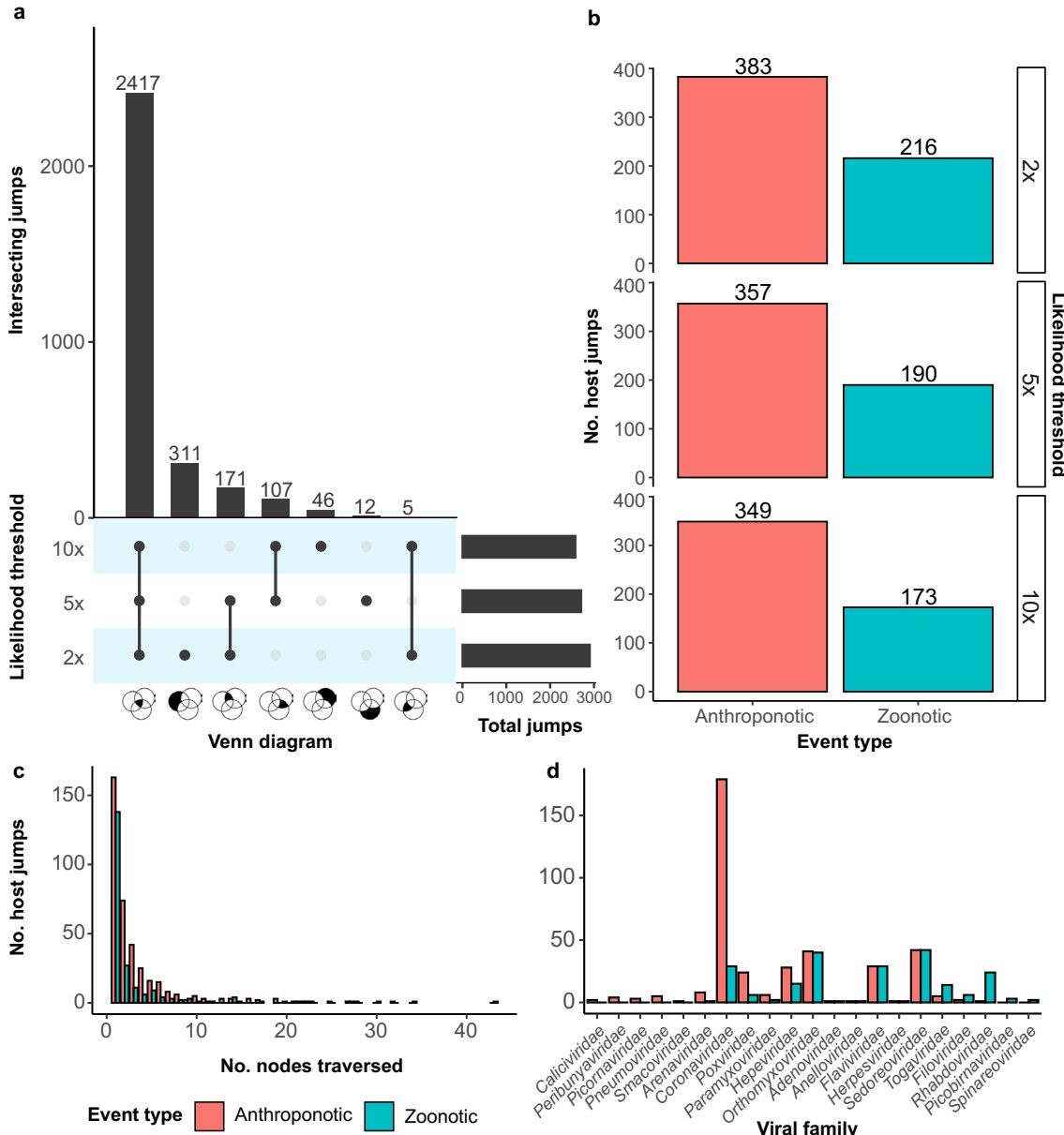

**Extended Data Fig. 5 | Robustness of host jump inference.** (**a**) UpSet plot providing the intersecting host jumps identified via ancestral reconstruction when using a two-fold, five-fold or ten-fold likelihood threshold. (**b**) Bar plot showing the number of anthroponotic and zoonotic events inferred using various likelihood thresholds, (**c**) at different ancestral node depths, and (**d**) stratified by viral family. For (**b**), the number of anthroponotic and zoonotic host jumps were stratified by the depth of the ancestral node in the tip-to-node traversal. Since multiple host jump lineages can involve the same ancestral node, the tip-to-node depths may vary depending on which lineage is selected. As such, we randomly selected a viral lineage for each distinct host jump event for this analysis.

**a**

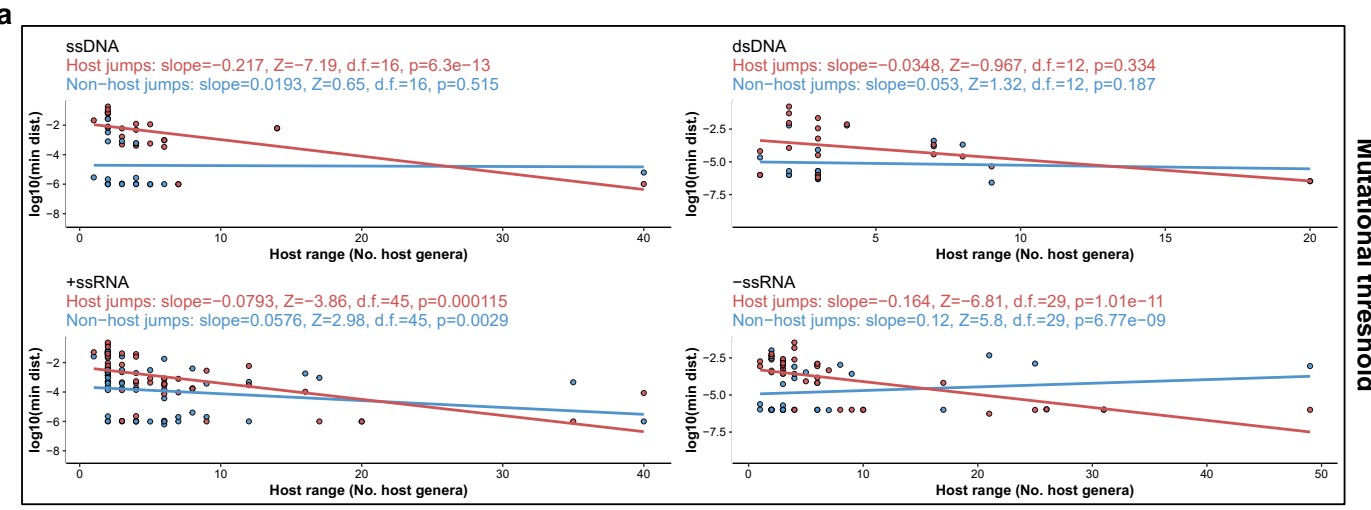

**b**

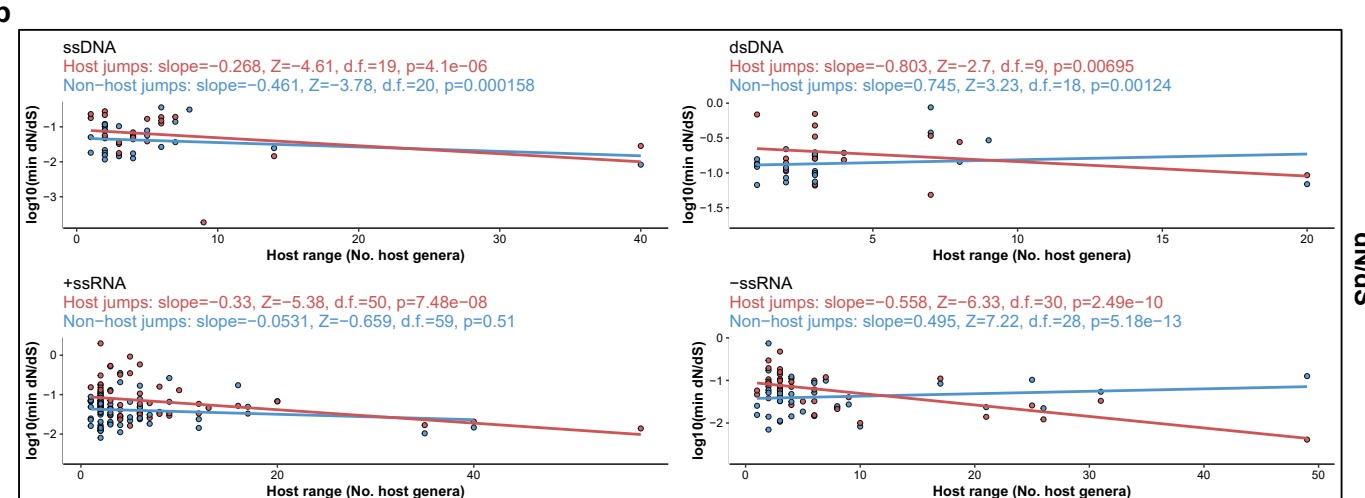

**Extended Data Fig. 6 | Adaptation analysis for viral groups.** Analysis of relationships between host range and estimated adaptive signals, similar to Fig. 3, but only considering ssDNA, dsDNA, +ssRNA or -ssRNA viruses. Distributions of minimum (**a**) mutational distance and (**b**) dN/dS for host jump and non-host jumps on the logarithmic scale. We corrected for the effects of sequencing effort and viral family membership using Poisson regression models. The estimated effects of patristic distance on host range after these corrections are annotated. We tested whether the estimated effects were non-zero using two-tailed Z-tests. For all panels, each data point represents the minimum distance or dN/dS across all host jump or randomly selected non-host jump lineages in a single clique. Line segments represent linear regression smooths without correction.

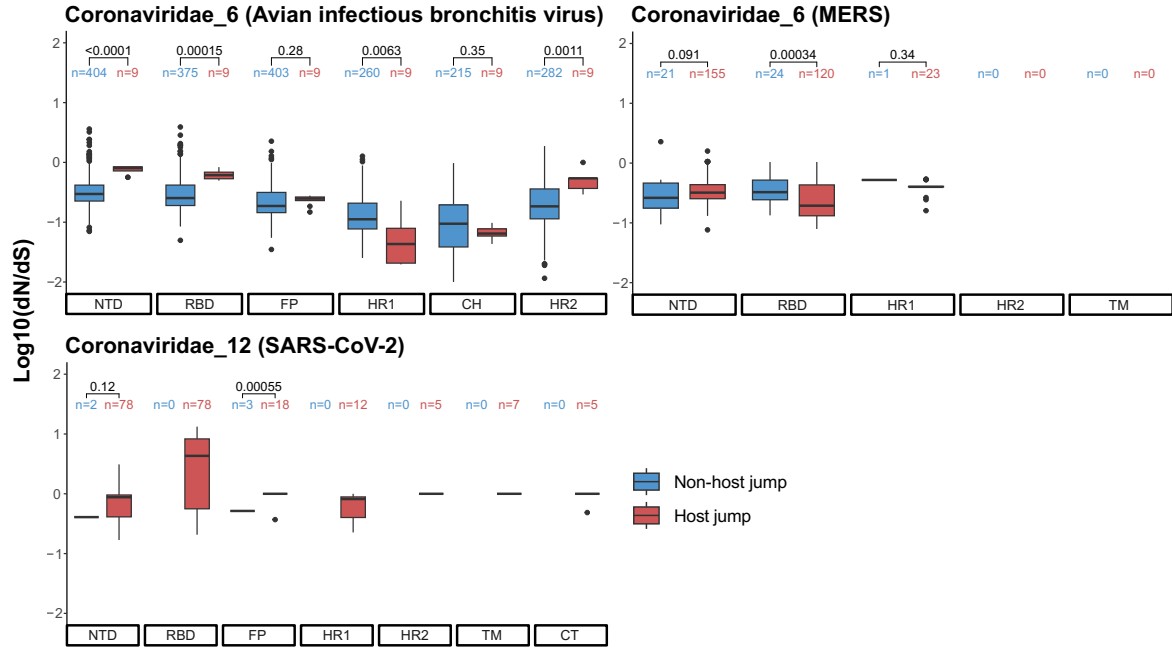

**Spike domain**

**Extended Data Fig. 7 | Adaptive signals in the Coronaviridae spike gene.**
Analysis of the log10(dN/dS) estimates associated to different functional
domains encoded by the coronavirus spike gene: N-terminal domain (NTD),
receptor-binding domain (RBD), fusion peptide (FP), heptad repeats 1 and 2
(HR1 and HR2), central helix (CH), transmembrane (TM), C-terminal domains
(CT). Estimates with dN=0 or dS=0 were removed and the remaining number
of sequences for each domain and viral clique are annotated. Differences in
distributions were tested for using two-sided Mann-Whitney U tests and the
corresponding p-values are annotated. Boxplot elements are defined as follows:
centre line, median; box limits, upper and lower quartiles; whiskers,
1.5x interquartile range.

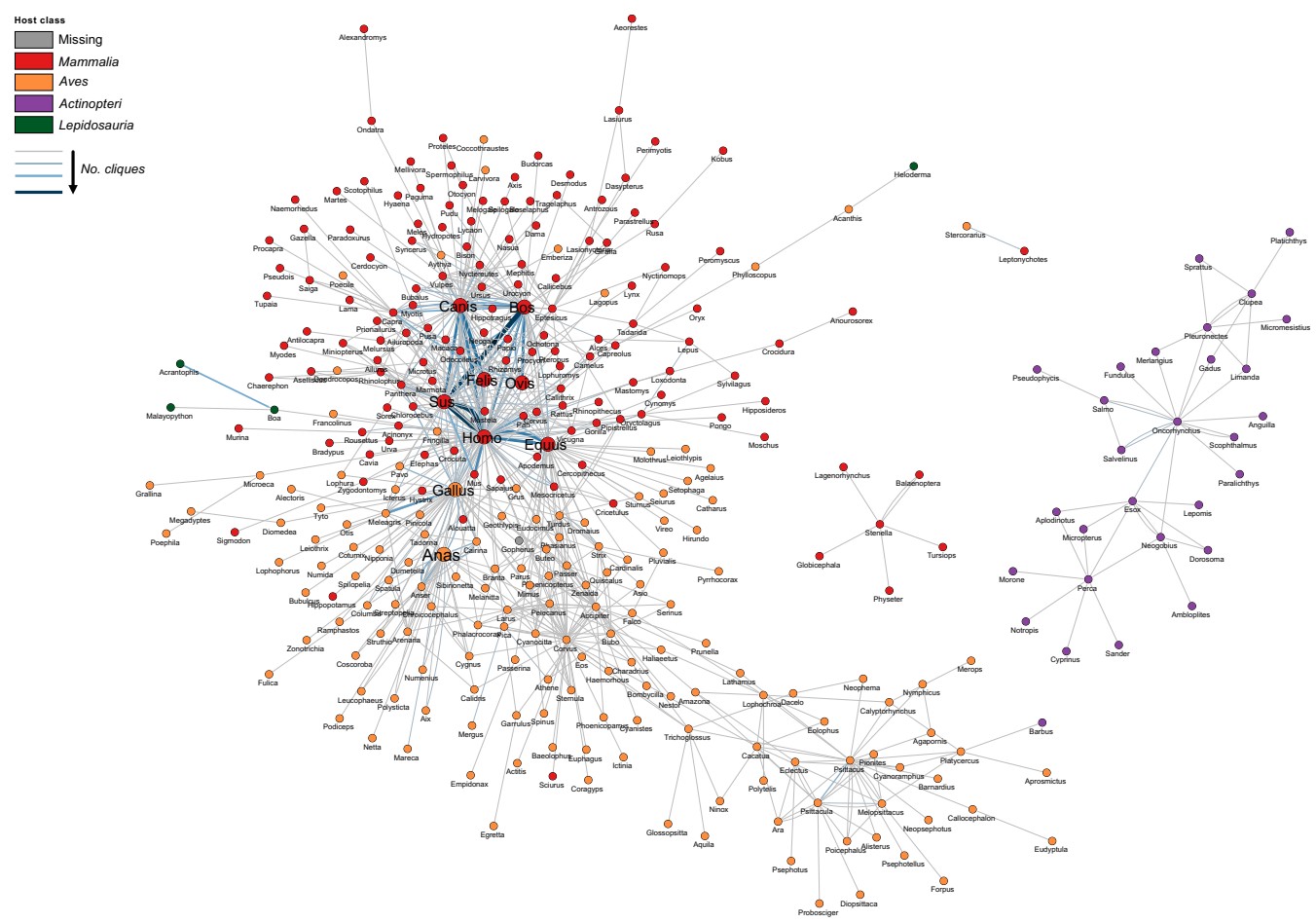

**Extended Data Fig. 8 | The global viral host jump network.** Directed network of the vertebrate viral-sharing network, where nodes and edges represent host genera and the number of viral cliques shared. Edge widths and colour are indicative of the number of viral cliques shared.

# Reporting Summary

## Statistics

For all statistical analyses, confirm that the following items are present in the figure legend, table legend, main text, or Methods section.

| n/a | Confirmed | |
|---|---|---|
| ☐ | ☒ | The exact sample size (*n*) for each experimental group/condition, given as a discrete number and unit of measurement |
| ☒ | ☐ | A statement on whether measurements were taken from distinct samples or whether the same sample was measured repeatedly |
| ☐ | ☒ | The statistical test(s) used AND whether they are one- or two-sided<br>*Only common tests should be described solely by name; describe more complex techniques in the Methods section.* |
| ☐ | ☒ | A description of all covariates tested |
| ☐ | ☒ | A description of any assumptions or corrections, such as tests of normality and adjustment for multiple comparisons |
| ☐ | ☒ | A full description of the statistical parameters including central tendency (e.g. means) or other basic estimates (e.g. regression coefficient) AND variation (e.g. standard deviation) or associated estimates of uncertainty (e.g. confidence intervals) |
| ☐ | ☒ | For null hypothesis testing, the test statistic (e.g. *F*, *t*, *r*) with confidence intervals, effect sizes, degrees of freedom and *P* value noted<br>*Give P values as exact values whenever suitable.* |
| ☒ | ☐ | For Bayesian analysis, information on the choice of priors and Markov chain Monte Carlo settings |
| ☒ | ☐ | For hierarchical and complex designs, identification of the appropriate level for tests and full reporting of outcomes |
| ☐ | ☒ | Estimates of effect sizes (e.g. Cohen's *d*, Pearson's *r*), indicating how they were calculated |

*Our web collection on statistics for biologists contains articles on many of the points above.*

## Software and code

Policy information about availability of computer code

| Data collection | taxizedb v0.3.1<br>ncbi-acc-download v0.2.8 |
|---|---|
| Data analysis | R v4.3.1<br>CheckV v1.0.1<br>Mash v1.1<br>Ape v5.7.1<br>Aricode v1.0.2<br>MAFFT v7.490<br>IQ-Tree v2.1.4-beta<br>Castor v1.7.10<br>MSA2dist v1.4.0<br>ggtree v3.8.2<br>ggplot v3.4.2 |

For manuscripts utilizing custom algorithms or software that are central to the research but not yet described in published literature, software must be made available to editors and reviewers. We strongly encourage code deposition in a community repository (e.g. GitHub). See the Nature Portfolio guidelines for submitting code & software for further information.

# Data

Policy information about availability of data

All manuscripts must include a data availability statement. This statement should provide the following information, where applicable:
- Accession codes, unique identifiers, or web links for publicly available datasets
- A description of any restrictions on data availability
- For clinical datasets or third party data, please ensure that the statement adheres to our policy

All custom code used to perform the analyses reported here are hosted on GitHub (https://github.com/cednotsed/vertebrate_host_jumps). The full list of accessions considered in this study are provided in Supplementary Table 1.

# Research involving human participants, their data, or biological material

Policy information about studies with human participants or human data. See also policy information about sex, gender (identity/presentation), and sexual orientation and race, ethnicity and racism.

| | |
|---|---|
| Reporting on sex and gender | N.A. |
| Reporting on race, ethnicity, or other socially relevant groupings | N.A. |
| Population characteristics | N.A. |
| Recruitment | N.A. |
| Ethics oversight | N.A. |

Note that full information on the approval of the study protocol must also be provided in the manuscript.

# Field-specific reporting

Please select the one below that is the best fit for your research. If you are not sure, read the appropriate sections before making your selection.

☒ Life sciences  ☐ Behavioural & social sciences  ☐ Ecological, evolutionary & environmental sciences

For a reference copy of the document with all sections, see nature.com/documents/nr-reporting-summary-flat.pdf

# Life sciences study design

All studies must disclose on these points even when the disclosure is negative.

| | |
|---|---|
| Sample size | No sample size calculation was performed. We used almost all publicly available genome sequences relevant to this study. Approximately 56k genomes were used for our analyses, and at least 500 genomes per group (i.e., viral family) were used, so sample sizes are sufficient. |
| Data exclusions | To generate a candidate list of viral sequences for further genomic analysis, the metadata was filtered to include 53 viral families known to infect vertebrate hosts based on information provided in the 2022 release of the ICTV taxonomy (https://ictv.global/taxonomy), and with reference to that provided by ViralZone (https://viralzone.expasy.org/). We then retained only sequences from viral families comprising at least 100 sequences of greater than 1000nt in length. For non-segmented genomes, we retained all non-human-associated sequences, and subsampled the human-associated sequences as follows: we selected a random subsample of 1000 SARS-CoV-2 genomes of greater than 28000nt from distinct countries, isolation sources, and with distinct collection dates. For other human-associated sequences, we retained viruses with distinct species, country, isolation source and collection date information. . We then downloaded the final candidate list of viral sequences (n=88,161) using the ncbi-acc-download v0.2.8 (https://github.com/kblin/ncbi-acc-download). Further quality control of the genomes downloaded was performed using CheckV v1.0.146, retaining sequences with more than 95% completeness (for non-segmented viruses) and less than 5% contamination (for all sequences). This resulted in a final genomic dataset comprising 53,631 observations (Supplementary Table 2).

For clique-level alignments, we masked regions of the alignments that were poorly aligned or prone to sequencing-error by replacing alignment sites that had more than 10% of gaps or ambiguous nucleotides with N's. Clique-level genome alignments that had more than 20% of the median genome length masked were considered to be poorly aligned and removed from further analysis (n=6; Extended Data Fig. 5) |
| Replication | No experimental findings were reported so this section is not applicable. |
| Randomization | Genomic datasets used for our study are retrospective and downloaded from public sequence databases so randomisation is not applicable to our study. |
| Blinding | Genomic datasets used for our study are retrospective and downloaded from public sequence databases so blinding is not applicable to our study. |

# Reporting for specific materials, systems and methods

We require information from authors about some types of materials, experimental systems and methods used in many studies. Here, indicate whether each material, system or method listed is relevant to your study. If you are not sure if a list item applies to your research, read the appropriate section before selecting a response.

## Materials & experimental systems

| n/a | Involved in the study |
|-----|-----------------------|
| ☒ ☐ | Antibodies |
| ☒ ☐ | Eukaryotic cell lines |
| ☒ ☐ | Palaeontology and archaeology |
| ☒ ☐ | Animals and other organisms |
| ☒ ☐ | Clinical data |
| ☒ ☐ | Dual use research of concern |
| ☒ ☐ | Plants |

## Methods

| n/a | Involved in the study |
|-----|-----------------------|
| ☒ ☐ | ChIP-seq |
| ☒ ☐ | Flow cytometry |
| ☒ ☐ | MRI-based neuroimaging |

